Metal to phosphorus stoichiometries for freshwater phytoplankton in three remote lakes

Gormley-Gallagher Aine M. a.m.gormley-gallagher@rug.nl
Douglas Richard W.
Rippey Brian
School of Geography & Environmental Sciences, University of Ulster , Coleraine , United Kingdom
Stanford Jack
Electronic publication date: 2016 Dec 20
Publication date: 2016
Volume: 4
Electronic Location ID: e2749
Received 2014 Nov 12; Accepted 2016 Nov 3
Copyright: ©2016 Gormley-Gallagher et al.
Copyright year: 2016
Copyright holder: Gormley-Gallagher et al.
License: This is an open access article distributed under the terms of the Creative Commons Attribution License, which permits unrestricted use, distribution, reproduction and adaptation in any medium and for any purpose provided that it is properly attributed. For attribution, the original author(s), title, publication source (PeerJ) and either DOI or URL of the article must be cited.
License URL: https://creativecommons.org/licenses/by/4.0/

Keywords: Metal, Phytoplankton, Phosphorus, Freshwater, Lakes, Stoichiometries, Scotland

Funding: Department for Employment and Learning (DEL) Northern Ireland This work was funded as a PhD studentship by the Department for Employment and Learning (DEL) Northern Ireland. The funders had no role in study design, data collection and analysis, decision to publish, or preparation of the manuscript.

==============================
Simultaneous measurements of changes in phytoplankton biomass and the metal and phosphorus (P) content of cells have been captured to attest to metal to P stoichiometries for freshwater phytoplankton. Three Scottish lakes that had received high, medium or low metal contamination from the atmosphere were selected for study. Phytoplankton cells were collected and Inductively Coupled Plasma-Mass Spectrometry was used to measure their lead (Pb), cadmium (Cd), mercury (Hg), copper (Cu), zinc (Zn), nickel (Ni), chromium (Cr), manganese (Mn), cobalt (Co) and P content. Increased phytoplankton growth in the lakes resulted in significant algae growth dilution of the mass-specific Pb, Cd, Hg, Cu, Ni and Cr in the phytoplankton. Changes in the phytoplankton cell count and their Hg, Pb, Cd, Cu, Mn, Co, Ni and Cr concentrations showed the process of algae bloom dilution to be subject to exponential decay, which accelerated in the order of Mn < Cu < Ni < Pb and Cd < Cr and Hg < Co. This indicated a metabolic and detoxification mechanism was involved in the active selection of metals. For the first time simultaneous measurements of metals and P stoichiometry in freshwater phytoplankton are reported. The mean metal to P stoichiometry generated was (C106P1N16)1000Pb0.019Hg0.00004Cu0.013Cd0.005Cr0.2Co0.0008Mn0.2Ni0.012 based on field measurements and the Redfield average C, N and P stoichiometry of (CH2O)106(NH3)16H3PO4.

Introduction

Phytoplankton cells are typically composed of carbon (C), nitrogen (N) and phosphorus (P) and have a commonly accepted average stoichiometry of (CH2O)106(NH3)16H3PO4 (Redfield, Ketchum & Richards, 1963; Sanudo-Wilhelmy et al., 2004). Phytoplankton can exploit iron (Fe), manganese (Mn), zinc (Zn), copper (Cu) and nickel (Ni) for N acquisition, oxygen cycling, chlorophyll synthesis, and sulfate reduction (Moffett et al., 1997; Twining, Baines & Fisher, 2004). These nutrient metals can be replaced at their metabolic site by micronutrient metals such as cadmium (Cd), mercury (Hg), lead (Pb) and chromium (Cr) (Bruland, Knauer & Martin, 1978; Sunda & Huntsman, 1998).

The cells can accumulate metals because they have a large surface area that has hydrophilic groups or hydroxy complexes with O-containing donor groups (-COH: -COOH; -P(O)(OH)2), which bind to ambient metal cations (Vasconcelos, Leal & Van den Berg, 2002). These sites on the cell surface are ligands from which metals can either dissociate back into solution or travel into the cytoplasm (Sunda & Huntsman, 1998). This has been reported as a dominant process of trace metal removal from solution (Whitfield, 2001; Lohan et al., 2005). Alternatively, cellular metal uptake may also occur through transport proteins or porins that are embedded in the outer membrane and allow for non-selective passive diffusion of metal ions across the outer membrane (Ma, Jacobsen & Giedroc, 2009).

Due to the realisation of the proclivity of metals to bind non-specifically to cell surfaces, studies have extended the concept of Redfield et al.’s nutrient stoichiometry to include metals. Ho et al. (2003) calculated a mean stoichiometry (mol:mol) of (C124N16P1S1.3K1.7Mg0.56Ca0.5)1000Sr5.0Fe7.5Zn0.80Cu0.38Co0.19Cd0.21Mo0.03,1 while Twining, Baines & Fisher (2004) found (C72P1S0.70)1000Zn5.4Fe1.8Ni0.61Mn0.26 for marine phytoplankton. Research has identified the risk posed to ecosystems and humans via toxic metal accumulation by phytoplankton with consequential transfer through the aquatic food chain (UNECE, 1998; Chen & Folt, 2000; Schultz & Seaward, 2000). As a result, the United Nations Economic Commission for Europe adopted the Heavy Metals Protocol to encourage modelling, research and descriptions of metal pathways (UNECE, 1998). Yet simultaneous measurements of metal to nutrient stoichiometry in freshwater phytoplankton have only been estimated (Wang & Dei, 2006).

When Reynolds & Hamilton-Taylor (1992) calculated stoichiometries of C106P1Zn0.034 for Lake Windermere, United Kingdom (UK), they estimated P based on regressions of dissolved P concentrations and the C: Si atomic ratio of 1:0.40 in phytoplankton cells. Likewise, Sigg (1985) and Sigg, Sturm & Kistler (1987) presented mean stoichiometries of C113P1Zn0.06Cu0.008 and (CH2O)106(NH3)16H3PO4Cu0.0006Zn0.03 for the phytoplankton of Lake Constance and Lake Zurich (Switzerland) respectively. However, the mean surface areas of the algae cells were estimated from correlation of the organic material content of the settling particles using typical cell dimensions of diatoms. Sigg therefore acknowledged that the stoichiometries are approximations that could vary if different algal species were taken into account.

Another mechanism (in addition to the influence of surface communities) that has been proposed to explain variations in metal to nutrient stoichiometries in phytoplankton is algae bloom density dilution. If phytoplankton share a finite pool of metals and have a constant uptake, enhanced lake productivity reduces metal concentrations per unit mass of phytoplankton (Chen & Folt, 2005). Additionally, if the trace element to macronutrient (i.e., phosphorus or carbon) ratios is a balance of net steady-state uptake and growth rates (Sunda & Huntsman, 1997; Sunda & Huntsman, 2004)—growth rates will increase as nutrients become more available, inducing a decline in cellular element to nutrient ratios. This suggests that, because P is a limiting nutrient for phytoplankton growth, increased cellular P would correlate with a decline in cellular metal concentrations.

Recommendations have been made that metal to P stoichiometries be incorporated into Biotic Ligand Models (BLM) (De Schamphelaere et al., 2005). When BLM were first developed, they provided a way to predict the ambient metal concentration that will have an effect (e.g., lethality) on organisms (e.g., fish), and emphasised the importance of including biological ligand concentration (e.g., physiologically active sites at the gills of fish) for that prediction (Di Toro et al., 2002). The models assumed a fixed rate of metal uptake occurred according to ambient concentrations, thus they were extended to include ambient water chemistry (Paquin et al., 2002). De Schamphelaere et al. (2005) then showed that cellular metal concentrations were better than ambient metal concentrations for predicting the threat of toxicity to freshwater phytoplankton. They stressed that cell surfaces should be used as the ligand for metals in the same way as fish gills apply to the BLM for predicting metal toxicity to fish species. Wang & Dei (2006) then showed that the metal to nutrient stoichiometry in phytoplankton cells better predicts metal toxicity than cellular metal burden. Determining what biogeochemical characteristics influence toxic metal uptake and accumulation in the aquatic food chain is important for identifying communities and species at risk of adverse impacts from metal contamination—and for developing management strategies to mitigate this risk (Ward et al., 2010).

The need for a simultaneous measurement of metal to nutrient (in this case P) stoichiometry in freshwater phytoplankton will be addressed in this contribution. Our underlying hypothesis is that cellular metal to P ratios decline as P becomes more available—and thus the dilution of metals in freshwater phytoplankton is a function of increased phytoplankton growth.

Figure 1 Regions of high, medium and low Pb contamination of lake sediment due to atmospheric deposition (from Rippey & Douglas, 2004).

The yellow circles in in the low, medium and high regions of contamination indicate the locations of Loch Coire nan Arr, Loch Doilet and Loch Urr, respectively (Gormley-Gallagher, Douglas & Rippey, 2015).

Site Descriptions

Investigations were undertaken in three UK lakes that have received varying degrees of metal contamination from the atmosphere (Rippey & Douglas, 2004). One lake was selected in each of the high, medium and low metal contaminated regions (Fig. 1). The three lakes receive metal contamination solely from atmospheric deposition—and thus metal contamination from runoff or direct discharges would not influence our results (Murray, 1987; Rippey & Douglas, 2004). Additionally, the size and bathymetry of the lakes meant that regular sediment resuspension events (and by association high suspended particulate matter) would be unlikely to influence our investigation outcome (Hilton, 1985; Douglas & Rippey, 2000; Douglas, Rippey & Gibson, 2003; Gormley-Gallagher, Douglas & Rippey, 2015).

In the following site descriptions, lake surface area, perimeter, altitude, grid reference, catchment area, maximum basin relief, and distance from the sea and to the nearest village were calculated and/or obtained using the OS Landranger® Memory-Map™V5 edition (2006) for northern and southern Scotland (Licence number PU 100034184). The maximum lake depths were based on collected field data, while catchment geology, vegetation and soil type were derived from Patrick, Monteith & Jenkins (1995).

Loch Coire nan Arr has a surface area of 13.21 ha, a maximum lake depth of 11 m and a catchment area of 8.45 km2 (Table 1). It is the most northerly of the three sites and lies in the region of low metal contamination from the atmosphere (Fig. 1). Permission for sampling the site was obtained from The Applecross Trust, a conservation charity responsible for the management of the lake (contact: admin@applecross.org.uk).

Loch Doilet has a surface are of 51.55 ha, a maximum lake depth of 16 m and a catchment area of 33.51 km2 (Table 1). The lake, lying northwest of the Ben Nevis Mountain range, is the largest of the three lakes and has received moderate metal contamination from the atmosphere (Fig. 1). The catchment rises from the lake to a peak of approximately 720 m. Permission for sampling the site was obtained from the Forestry Commission Scotland, a UK non-ministerial government department responsible for the management of the lake (contact: lochaber@forestry.gsi.gov.uk).

Table 1 Summary of the site characteristics of Loch Coire nan Arr in northwestern Scotland, Loch Doilet in western Scotland and Loch Urr in southern Scotland (Gormley-Gallagher, Douglas & Rippey, 2015).

	Loch Coire nan Arr	Loch Doilet	Loch Urr	
Grid Reference	NG808422	NM807677	NX759864	
Surface area	13.21 ha	51.55 ha	47.0 ha	
Perimeter	1.86 km	5.49 km	4.2 km	
Maximum lake depth	11 m	16 m	13.2 m	
Lake volume	5.6 ×105m3	4.1 ×106m3	2.35 ×106m3	
Distance upstream from sea	2.03 km	6.2 km	22.7 km	
Aerial distance from nearest village	8.91 km (Lochcarron)	8.84 km (Strontian)	6.6 km (Monaive)	
Elevation/altitude	125 m	8 m	193 m	
Catchment area	8.45 km2	33.51 km2	7.73 km2	
Catchment geology	Torridonian Sandstone	Schists and gneiss	Granite / gneiss	
Catchment vegetation	Confiers < 1%	Conifers—50%, moorland—50%	Moorland—100%	
Catchment soils	Peat	Peats	Podsol, peaty gley blanket peat	

Loch Urr has a surface area of 47 ha with a maximum lake depth of 13 m (Table 1). It lies in the Dumfries and Galloway region of south-west Scotland, an area that has received high metal contamination from the atmosphere (Fig. 1). The lake drains the smallest of the three catchments with an area of 7.73 km2. Permission for sampling the site was obtained from the Urr District Salmon Fisheries Board (contact: mail@gallowayfisheriestrust.org).

Sampling

Sampling campaigns were conducted on ten occasions: early June, late June, July, August and September 2006, and again in March, May, June, July and September 2007 at each of the three lakes. Before fieldwork, all sample containers were prepared to reduce metal contamination and prevent adsorption losses to the container walls (Yu et al., 2003).

During fieldwork, three lake water samples were collected from each lake. The first sample was for the analysis of chlorophyll-a, total phosphorus (TP) and pH. The second was for analysis of total metal concentrations. The third was for phytoplankton identification and calculations of biomass. The water was taken from a central location (6 m) near the deepest point of the lake using a Perspex Ruttner sampler, as recommended by Sykes, Lane & George (1999).

Phytoplankton samples were also collected from the lakes on each of the sampling occasions using the net haul method (Vollenweider, 1974). A 20 µm mesh net (30 cm wide) was used (EB Nets, UK) to take 10–18 hauls (varying with lake productivity) to concentrate phytoplankton. An adjustment was made to the standard nets to separate the zooplankton during each haul using the approach set out by Ho et al. (2007). Two filters, one of 20 µm and one of 250 µm were stacked on top of each other with a 35 mm spacer such that water flowed first through the 250 µm and then the 20 µm filter. The upper filter of mesh 250  µm was a sufficient size to trap the zooplankton but allow the smaller phytoplankton to be trapped in the smaller 20 µm mesh. This method potentially introduces sources of error. Firstly, by excluding bacterioplankton (free floating bacterial component of the plankton) and phytoplankton <20 µm from the metals estimate, the relationships of phytoplankton metal concentrations and TP could be affected. Secondly, possible clogging in the larger size fraction could lead to the selection of smaller phytoplankton in the sample. However, when the two size fractions were microscopically analysed, the zooplankton were not incorporated into the phytoplankton samples and phytoplankton smaller than 250 µm were not observed in the larger fraction. This may be attributed to the fact that zooplankton production and the concentration of suspended particulate matter are generally low in the lakes (Monteith & Shilland, 2007; Murphy et al., 2014; Gormley-Gallagher, Douglas & Rippey, 2015). The success of this method has also been demonstrated by Donald (2004) and in the larger study from which this investigation stems (Gormley, 2008). Furthermore, given the sampling difficulty in collecting sufficient uncontaminated biomass for metal analysis, abating sample handling by separating the plankton assemblages in this manner in-situ was deemed critical to minimise the possibility of metal contamination (Ho et al., 2007).

The water samples collected for phytoplankton identification and biomass calculations were transferred on site from LDPE bottles to acid washed scintillation vials (25 ml) that were pre-prepared with glutaraldehyde (Electron Microscopy grade, EMS, Pennsylvania, USA) to produce a final concentration of 2% (v/v).

The net haul material was transferred to a total of 36 polyethylene acid cleaned sampling vials (32 ml) at each site (AGB Scientific Ltd., UK). The vials used to store the net haul material were also pre-prepared to achieve 2% glutaraldehyde in the sample, except in this case, the glutaraldehyde was passed through a Dowex 50-W X8-200 cation exchange resin (50X4-400; H-form) to remove trace metals (Twining, Baines & Fisher, 2004).

Sample Analysis

TP concentrations were measured spectrometrically after digestions at 882 nm (Murphy & Riley, 1962; Eisenreich, Bannerman & Armstrong, 1975). Chlorophyll-a was extracted from the filtered samples into 90% V/V methanol, and the detection was performed with a spectrophotometer set at an emission wavelength of 665 nm (Riemann, 1978). A Shimadzu UV-Mini 1240 Spectrophotometer was used for this at the Ulster University.

A Nikon-5400 inverted light microscope at 40× was used to examine the phytoplankton samples and identify the species present. For this, 10 ml of the lake water sample preserved in glutaraldehyde was allowed to sediment in a settling chamber for no less than 8 h. Blue–green and green algae organisms were identified following the interactive keys produced by Whitton et al. (2002) and Whitton, Balbi & Donaldson (2003). For those organisms that proved difficult to distinguish, a more detailed text was consulted, i.e., John, Whitton & Brook (2002). The guidelines presented by Kelly (2000) were followed to identify any cells representative of the Phylum Bacillariophyta and the Phylum Fragilariophyceae (Diatoms).

During identification, the species/genre/groups were also counted and measured for volume and surface area calculations following the procedures described by Olrik et al. (1998). At least 10 length and width measurements were recorded for each species (wall to wall), and when fewer than 10 cells were present, those present were measured. Cell counts were converted to counts per volume of lake water. Cell volumes and surface areas were calculated using the geometric equations of Hillebrand et al. (1999). The volume of colonial and filamentous cells was calculated from the volume of a single cell multiplied by the number of cells in each colony/filament. The surface area of cells per volume of lake water was then calculated following the guidelines of Olrik et al. (1998).

Acid digestions were prepared using methods found in Reynolds & Hamilton-Taylor (1992). To achieve blank concentrations, 2 × 32 ml vials of 2% glutaraldehyde were prepared prior to each fieldwork session and brought on fieldwork to ensure they had the same sample exposure. On return to the laboratory, a stream of Milli-Q water was used to fill the vial as it was passed through the same plankton net filter used to collect the samples.

The phytoplankton samples were made soluble (digested) by treatment with hydrofluoric, nitric and perchloric acid, following the acid digestion technique provided in Bock (1979). An empty beaker (a reagent blank), and two samples of certified reference material (CRM) were included with every batch (between 20–30 samples). The CRM used for this study was Chinese stream sediment (GBW 07301) issued under the laboratory of the Government Chemist (LGC) trademark (LGC Promochem, UK). The digested samples were stored in acid cleaned 25 ml scintillation vials until further analysis with Inductively Coupled Plasma-Mass Spectrometry (ICP-MS).

The XSeriesI ICP-MS (ThermoFisher Scientific Cooperation) was used for the analysis of metals and P in the samples (Table S1). All prepared standard solutions, samples and blanks were acidified with 2% (w/v) HNO3− (BDH Aristar, AGB Scientific Ltd., UK). The precision of every element was assessed from replicate and, when possible, triplicate analysis of reference material and of samples collected in fieldwork. This was found to be 5% relative standard deviation (RSD) or better, which is generally considered acceptable precision (Long, Martin & Martin, 1990). Also, instrument stability was indicated in the RSD of triplicate ICP-MS measurements for all analytes of less than 5% in all cases, and in many cases less than 2%.

Figure 2 Chlorophyll-a (Chl-a) and total phosphorus (TP) concentrations measured in Loch Coire nan Arr (A), Loch Doilet (B), and Loch Urr (C).

The series keys located in the top left of the diagram applies to each of the trend lines. Error bars are the standard error between the triplicate measurements of each result (n = 3).

Figure 3 Concentrations of Pb, Hg, Cd, Cu, Cr, Co, Mn and P determined per unit mass of the phytoplankton cells collected in Loch Coire nan Arr, Loch Doilet, and Loch Urr.

Concentrations of Hg, Cd, Cu and Co determined per unit mass of the phytoplankton cells collected in Loch Coire nan Arr (A), Loch Doilet (B), and Loch Urr (C). Concentrations of Mn, P, Cr and Pb determined per unit mass of the phytoplankton cells collected in Loch Coire nan Arr (D), Loch Doilet (E), and Loch Urr (F). All values are in µg of metal per g of phytoplankton. The series key located in the (A) diagram applies to A–C. The series key located in the (D) diagram applies to D–F.

Results

The measured concentrations of chlorophyll-a and TP and modelled chlorophyll-a concentrations based on OECD (1982) and Prairie, Duarte & Kalff (1989) models for predicting chlorophyll-a based on TP concentrations are presented in Fig. 2. The peak of TP concentrations was recorded in mid-May 2007 for Loch Doilet (23.5 µg/l) and Coire nan Arr (79.3 µg/l), whereas the peak in Loch Urr (85.3 µg/l) occurred in late September 2006. The chlorophyll-a trends in Fig. 2 show a peak during August/September 2006 for Loch Doilet (3.10 µg/l) and September 2007 for Loch Urr (23.0 µg/l), whereas the peak in Loch Coire nan Arr was during the month of July 2007 (10.25 µg/l). The lowest chlorophyll-a concentrations were 1.4, 1.5 and 2.7 µg/l respectively for Loch Coire nan Arr, Loch Doilet and Loch Urr. In many cases, Fig. 2 shows that an increase in TP is followed by a rise in chlorophyll-a on the subsequent sampling occasion, particularly in Loch Coire nan Arr and Loch Urr. Also, the patterns of chlorophyll-a generally show similar timing in their fluctuations to that of the predictions of chlorophyll-a concentrations, notably in Loch Doilet.

Figure 4 Correlation between Pb, Cd, Hg, Cr, Cu and Ni concentrations per unit mass of phytoplankton and TP concentrations.

Correlation between Pb (A), Hg (B), Cu (C), Cd (D), Cr (E) and Ni (F) concentrations per unit mass of phytoplankton and TP concentrations. The data was collected from the samples of all three lakes during each sampling occasion (n = 29).

Figure 3 shows the concentrations of Pb, Hg, Cd, Cu, Cr, Co, Mn and P determined per unit mass of the phytoplankton cells in Loch Coire nan Arr, Loch Doilet and Loch Urr. The trend lines show high fluctuation across the sampling dates from early June 2006 to September 2007. In the majority of cases the phytoplankton of Loch Urr held the lowest concentrations of metals, but the highest concentration of P in the cells.

The concentration of Pb, Cd, Hg, Cr, Cu and Ni per unit mass of phytoplankton cells is plotted against the TP concentrations of the three lakes on all sampling occasions in Figure 4 (n = 29). The scatterplots show a linear relationship with negative slope between each of the two sets of variables. This indicates that the lower the lake TP concentration, the higher the concentration of metals per unit mass of phytoplankton. Before completing the regression analysis in Fig. 4, the Kolmogorov–Smirnov and Shapiro–Wilk’s tests on the normality of the (raw) data showed the TP concentrations and the mass-specific metal concentration in the phytoplankton to not be normally distributed (p < 0.05). However, using the log-transformed metal concentrations and the cubic root of TP concentrations, the data showed normal distribution (p > 0.05) in the Kolmogorov–Smirnov and Shapiro–Wilk’s tests.

A bivariate correlation and regression analysis was carried out on the data in Fig. 4 using the Statistical Package for Social Science (SPSS). The correlation coefficient and p-values of the tests confirms the patterns in the scatterplot that a significant negative relationship exists between TP and Pb (r =  − 0.823, p = 0.00), Hg (r =  − 0.741, p = 0.01), Cu (r =  − 0.748, p = 0.00), Cd (r =  − 0.662, p = 0.00), Cr (r =  − 0.837, p = 0.00) and Ni (r =  − 0.532, p = 0.02) per unit mass of phytoplankton in the lakes.

In contrast to Pb, Cd, Hg, Cu, Cr and Ni, Co, Mn and P per unit mass of phytoplankton cells showed no clear relationship against the TP concentrations of the three lakes on all sampling occasions. Examination of the bivariate correlation between the variables indicated no significant relationship exists. Due to the extensive number of outliers and the lack of significant correlation between the two sets of variables, a regression analysis was not suitable for the data.

Table 2 Summary of the simultaneous multiple regression performed using chlorophyll-a and total phosphorus (TP) as independent variables and the metal (Pb, Cd, Cr, Hg, Cu, Mn, Co) to P ratios in phytoplankton cells from the three lakes as the dependant variable.

Where p < 0.05, the relationship was significant at the 5% level, and where p < 0.10, the relationship is significant at the 10% level.

Metal	Metal : P ratio with	
	Chlorophyll a	Total phosphorus	
	t	Sig.	t	Sig.	
Pb	−0.474	0.640	−2.541	0.017	
Cd	−0.179	0.859	−2.457	0.021	
Cr	−0.384	0.704	−2.781	0.010	
Hg	−1.018	0.318	−1.710	0.099	
Cu	−0.507	0.616	−1.189	0.245	
Mn	0.167	0.896	0.683	0.501	
Co	−0.635	0.531	0.187	0.853	

Figure 5 The relationship between TP and the Pb (A), Hg (B), Cd (C), Cr (D) to P ratios per unit mass of phytoplankton cells in the three lakes.

As a single variable in the multiple regression between the metal: P ratios against chlorophyll-a and TP, TP is a significant predictor of Pb, Cd and Cr: P ratios at the 5% level, and of Hg: P at the 10% level (Table 2).

Table 2 summarises the results of the multiple regressions carried out using a combination of chlorophyll-a and TP (as the independent variables) against metal (Pb, Cd, Cr, Hg, Cu, Mn, Co) to P ratios per unit mass of phytoplankton cells (the dependant variable). An examination of the t-values in Table 2 indicates that TP is a significant predictor of the variations in Pb:P, Cd:P and Cr:P ratios in cells at the 5% level, but chlorophyll a alone is not. For the Hg:P ratio in cells, TP is a significant predictor at the 10% level, but chlorophyll-a alone is not a significant predictor.

The relationships in Table 2 are illustrated in Fig. 5. This shows the strongest correlation to exist between the Cr:P ratio in cells and TP (r2 = 0.3362).

Figure 6 The dominant groups of phytoplankton (as a percentage of the total volume) in the three lakes.

The percentage composition is presented for Loch Coire nan Arr (A), Loch Doilet (B) and Loch Urr (C). The series key located in diagram (A) applies to (A–C).

Figure 6 shows the dominant groups of phytoplankton (as a percentage of the total volume), illustrating the shifts in species association of the phytoplankton over the sampling period. Among these, the dominant groups in Loch Coire nan Arr (Fig. 6A) were the Chlorophytes (particularly Cosmarium sp.) and the Dinoflagellates (particularly Peridinium willei). In Loch Doilet (Fig. 6B), the Chlorophytes were also a dominant group, particularly the filament Oedogonium sp. In contrast, Loch Urr (Fig. 6C) had a greater abundance of the blue–green algae, such as the genus Oscillatoria sp., which is from the prokaryotic group the Cyanophytes. There was also a higher dominance of the Diatoms in Loch Urr in comparison to the other lakes.

Table 3 Biomass, surface area and cell count determined for the three lakes on each of the sampling occasions.

Lake	Date	Biomass (µg/l)	Surface area (mm2/l)	Cell count (no./ml)	
Loch Coire nan Arr	22.09.06	4.4	3.1	4.7	
	23.03.07	7.1	7.1	8.9	
	22.05.07	12.9	9.6	33.9	
	21.06.07	77.5	23.1	52.5	
	25.07.07	1.4	1.7	8.9	
	01.09.07	10.1	14.8	39.8	
Loch Doilet	22.09.06	2.9	3.7	4.4	
	23.03.07	0.5	1.4	12.4	
	22.05.07	3.8	2.3	8.0	
	21.06.07	17.5	3.8	1.2	
	24.07.07	1.8	1.4	5.5	
	31.08.07	35.7	34.6	31.7	
Loch Urr	21.09.06	63.6	3564.3	197.6	
	23.03.07	169.8	9278.3	172.3	
	23.05.07	178.2	138.9	183.3	
	22.06.07	69.3	48.6	99.2	
	26.07.07	115.0	1804.8	307.5	
	02.09.07	445.6	263.9	278.0	

Figure 7 Correlation between cell count and TP concentrations from early June 2006 to late September 2007 in all three lakes.

The significance (p) value was computed with SPSS on the significance of the regression line.

The biomass, surface area and cell count calculated for Loch Coire nan Arr, Loch Doilet, and Loch Urr are detailed in Table 3. Based on these data, the correlation between cells count and TP at the time of sampling was significant at the 5% level (Fig. 7), however, the correlations between TP and cell surface area as well as biomass were not significant at the 5% level.

The regression models obtained for TP and cell count (Fig. 7), and those generated for cell count and the concentration of metals per gram of cells (Eqs. 1–8) were used to calculate the best-fit values that describe the effect of changes in cell density on metal uptake by the phytoplankton under different trophic states.

This was completed by firstly using the regression equation for TP and cell count (Fig. 7) to estimate the number of cells per ml under a range of TP concentrations. These data were then incorporated into the following regression equations obtained from the analysis of the metals and P per unit mass of phytoplankton and the corresponding cell count. (1) Pb=−1.888× log10cell count+9.9733

(2) Hg=−0.268× log10cell count+1.3543

(3) Cu=−0.874× log10cell count+5.8133

(4) Cd=−1.006× log10cell count+4.8643

(5) Cr=−2.530× log10cell count+13.4123

(6) Co=−0.538× log10cell count+2.5723

(7) Mn=−0.967× log10cell count+10.6093

(8) P=−1.114× log10cell count+16.5513

This generated best-fit values for each metal per gram of cells. For example, the Hg per gram of phytoplankton in water with a TP concentration of 30 µg/l was calculated as follows:

• Phytoplankton cells per ml:

=10 ˆ  ((0.7188 × 301∕3) – 0.727)

=31.9 cells.

• Hg per gram of phytoplankton:

= ((−0.268 × log (31.9)) + 1.354)3

=0.86 µg/g.

Table 4 provides details on how the predicted Hg concentrations change per gram of cells with a range of TP concentrations.

Table 4 Best-fit values of the number of phytoplankton cells per ml under a range of trophic states and the concentration of Hg per unit mass of those cells.

The cells per ml were predicted using the regression formula generated for TP and cell counts in this study (Fig. 7). Concentrations of Hg per µg of cells were estimated using the predicted cells per ml and the regression equation for Hg per unit mass of phytoplankton (Eq. 2).

TP (µg/l)	Phytoplankton cells per ml	Hg per unit mass of cells (µg/g)	
10	6.61	1.46	
12	8.27	1.36	
14	10.09	1.28	
16	12.09	1.20	
18	14.29	1.14	
20	16.68	1.08	
22	19.28	1.03	
24	22.09	0.98	
26	25.13	0.94	
28	28.40	0.90	
30	31.91	0.86	

Figure 8 shows the best-fit lines for the relationship of cell counts and the concentration of Hg, Cd, Cr, Cu, Co, Mn, Ni and Pb per gram of cells. These were calculated in the same way as described in detail for Hg, with an extension of that data to include the range of TP values recorded in this study (7–85 µg/l). As the best-fit curves are without noise, and because they represent the correlations in the data obtained from this study, they can be used to examine the rate of metal uptake by phytoplankton cells in this study. The data points, i.e., the true measurements recorded, were used in an exponential regression to quantitatively describe the rate of uptake by the phytoplankton.

The best-fit lines in Fig. 8 suggest that the uptake of Hg, Pb, Cd, Cu, Co, Ni and Cr by the phytoplankton is subject to exponential decay. This is characterised by an initially rapid decline in metal concentrations per µg of phytoplankton with increasing cells, until the concentration approaches zero, where the rate of the absolute decrease in the metals decelerates. The exponential regression equations for the data points in Fig. 8 shows the decay constant, which defines the rate of metal decay in phytoplankton cells with an increasing number of cells. The larger the rate constant, the more rapid the decay of the dependent variable (y, metals in phytoplankton). The rate of Pb, Cd, Cr, Hg, Cu, Co, Ni and Mn decay in phytoplankton cells with an increasing number of cells is 0.0046, 0.0046, 0.0045, 0.0045, 0.0037, 0.0069, 0.004 and 0.0031 (mL/cell) respectively.

Figure 8 Relationship of phytoplankton cell counts with Pb (A), Cr (B), Cu (C), Mn (D), Cd (E), Hg (F), Co (G) and Ni (H) per gram of cells.

The best-fit lines (in red) were calculated from the predicted cell counts (Fig. 7) and the metal (and P) concentrations per unit mass of cells (Fig. 3). The data points are the actual measurements recorded in this study and were used in the exponential regression of the formula displayed for each relationship.

Figure 9 The relationship of phytoplankton cell counts with TP concentrations.

The best-fit line was calculated from the regression analysis of TP and cell counts (Fig. 7). The data points are the actual measurements recorded in this study and were used in the exponential regression of the formula displayed.

As an additional observation, Fig. 9 shows the line of best-fit for TP and phytoplankton cell count. This was calculated with the regression models obtained for TP and cell count (Fig. 7). The data points are the actual measurements recorded, and were used for the exponential regression analysis displayed to quantitatively describe the growth of cells in response to rising TP conditions. Figure 9 suggests that cell production with increased TP concentrations is subject to exponential growth. This is characterised by an initial gradual rise in cell count with increasing TP, but as more TP is introduced, the rate of growth accelerates.

The metal concentrations in one cell of phytoplankton were calculated by firstly calculating the weight of an individual cell. For example, in Loch Doilet on the 22/05/2007 the phytoplankton cell count was 7.95 cells/ml and the mean phytoplankton biomass was 3.77 µg/l (Table 3). Therefore the weight of one cell is calculated as follows.

• Phytoplankton cell biomass (µg/l) ÷ number of cells per litre (cells/l)

=3.77 µg/l ÷ 7950 cells/l

=4.74 × 10−4µg (mean weight of one cell in Loch Doilet).

Secondly, the concentration of metals was calculated for one cell. This was carried out by using the weight of one cell and the concentration of metal per unit weight of cells. The above cell weight for Loch Doilet on the 22/03/2007 and the concentration of Cd per gram of cells will be used as an example here.

• Weight of individual cell (g/cell) × Cd per gram of cells (µg/g)

=4.74 × 10−10(g/cell) × 8.5 (µg/g)

=4.03 × 10−15g of Cd per cell.

Table 5 shows the calculated concentrations for Hg, Pb, Cd, Cu, Cr, Co, P, Mn and Ni in the phytoplankton cells of each of the lakes on all sampling occasions.

Table 5 Content of Pb, Cd, Hg, Cr, Co, Ni, Mn, P and Cu per phytoplankton cell in the three lakes on all sampling occasions.

The values were calculated from the average weight of one cell, and the metal (and P) concentrations per gram of cell on the same date.

Lake	Date	Metal content per phytoplankton cell	
		Pb (g ×10−15)	Cd (g ×10−15)	Hg (g ×10−15)	Cr (g ×10−14)	Co (g ×10−16)	Ni (g ×10−14)	Mn (g ×10−14)	P (g ×10−12)	Cu (g ×10−14)	
Loch Coire nan Arr	22.09.06	906.6	139.0	1.5	207.5	77.3	13.2	138.2	1.8	12.6	
23.03.07	1444.3	200.1	2.2	325.3	124.1	20.4	333.6	3.2	20.4	
22.05.07	14.5	0.4	0.1	3.0	16.4	1.0	14.8	0.2	0.6	
21.06.07	343.7	44.4	0.3	68.2	97.5	2.5	137.9	6.0	6.7	
25.07.07	90.2	12.5	0.1	20.0	6.4	0.5	3.3	0.7	1.3	
01.09.07	229.8	41.4	0.2	53.3	8.4	0.7	5.9	2.1	3.3	
Loch Doilet	22.09.06	471.3	19.1	3.6	125.9	693.9	40.7	520.8	2.8	24.1	
23.03.07	4.3	0.1	0.0	1.2	4.7	0.4	2.1	0.1	0.3	
22.05.07	124.1	4.0	1.2	45.4	46.5	19.4	34.7	0.9	9.0	
21.06.07	9461.8	1509.8	16.9	2255.6	434.5	54.6	479.5	100.7	152.5	
24.07.07	273.6	44.6	0.5	66.6	7.5	1.7	6.6	2.1	4.3	
31.08.07	797.7	116.5	1.1	189.6	33.8	2.2	27.9	6.8	11.6	
Loch Urr	21.09.06	27.5	2.0	0.2	8.0	9.3	2.8	54.1	1.1	2.0	
23.03.07	92.4	3.2	0.6	29.9	32.5	10.8	113.1	1.7	6.5	
23.05.07	195.7	4.9	0.9	52.8	103.0	19.6	559.3	1.4	19.4	
22.06.07	230.4	31.9	0.4	52.1	5.6	1.3	4.1	3.7	3.1	
26.07.07	80.7	12.5	0.1	19.0	3.0	0.5	4.0	1.2	1.2	
02.09.07	333.6	46.9	0.4	76.7	9.6	2.0	11.9	4.0	4.9	

Discussion

As P is a limiting nutrient for phytoplankton growth, TP is a good measure of a lakes trophic status (Brooks, Bennion & Birks, 2001). From the range (maximum to minimum) of TP concentrations recorded for each lake (Fig. 2), the associated trophic status of the lakes ranges from oligio- mesotropic for Loch Doilet (3.7–23.5 µg TP l−1), oligio- eutrophic for Loch Coire nan Arr (2.7–79.3 µg TP l−1), and meso- eutrophic for Loch Urr (22.0–85.3 µg TP l−1). However, the trophic state of a lake is often judged in terms of mean TP concentrations (Carlson, 1977; Knowlton & Jones, 1997; O’Gorman, Lantry & Schneider, 2004). If the mean TP concentrations over the sampling period are used to assign a trophic status to the lakes in this study, that yields a status of mesotrophic for Loch Coire nan Arr with a mean TP of 22.9 µg/l, oligotrophic for Loch Doilet (9.6 µg TP l−1), and eutrophic for Loch Urr (45.9 µg TP l−1). The variation in the mean trophic state between the three lakes may be partially attributed to several differences in lake and catchment morphometry. For example, Loch Doilet has the lowest mean TP concentration at 9.6 µg TP l−1 but has a lake volume (4.2 × 106 m3) that greatly exceeds that of the other two lakes (5.0 × 105 m3 in Loch Coire nan Arr, 2.4 × 106 m3 in Loch Urr). It also has a relatively higher maximum lake depth recorded at approximately 16 m in comparison to a maximum depth of 12 m recorded in the other two lakes (Table 1). A larger lake volume and maximum depth tends to result in lower nutrient concentrations (Chow-Fraser, 1991). This is because firstly, a high volume of lake water can dilute the TP, and secondly, at greater lake depths there is less possibility of mixing and therefore P can be more readily removed from the water column by the sediment to the lake bed (Jeppesen et al., 2003).

The correlation between TP and the number of cells per ml at the time of sampling was significant at the 5% level (Fig. 7), however the correlations between TP and cell surface area as well as biomass were not. Insignificance in the correlation of TP and surface area has been previously noted by Thomann (1977) who suggests that the relationship is a combination of biomass, TP, retention time, and sinking rates. It is possible that the measurements of phytoplankton cell count, surface area and biomass in this study responded to TP at different rates. For example, count can remain constant even if volume increases, but if the volume per cell declines then the opposite applies, i.e., cell total volume remains constant but the number of cells increases. Surface area can vary with either, for example, a small spherical cell can have a greater surface area to volume ratio than a larger spherical cell. Equally, the variations in the correlations may also be because the method for the determination of cell count is open to less error than that of cell surface area and/or biomass. The latter are an extension of the determination of cell count and their final values include measurements of cell dimensions that fit into an assigned geometric formula. Additionally, Gleskes & Kraay (1983) and Reyonlds (1984) shed doubt on the accuracy of the ‘classical method’ for the quantification of phytoplankton growth. This is because it is based on spot samples that do not account for lateral and vertical fluctuations in lake temperature, nutrients and light availability, as these strongly influence the species composition and abundance of phytoplankton. Phycologists have also recognised that phytoplankton biomass can never be accurately quantified due to diurnal variations (B Whitton, pers. comm., 2006). Furthermore, the abundance of bacterioplankton and phytoplankton <20 µm are not accounted for in this investigation. As the bacterioplankton and phytoplankton <20 µm can compete with algae for P in the water column (Currie , 1990), a rise in TP concentrations in the samples analysed may not be accompanied by a rise in phytoplankton growth indicators in another sample from that same environment. Considering the significant relationship between TP and cell count, and that the use of cell count introduces the least error to the final result, it is perhaps more accurate to base interpretations of phytoplankton growth and metal interactions on cell count as opposed to biomass or surface area.

The significant correlations between the mass-specific Pb, Cd, Hg, Cr, Cu and Ni in the phytoplankton and TP concentrations (Fig. 4) suggest that algae bloom density dilution occurred in the lakes investigated. This evidence supports the findings of Pickhardt et al. (2002) for algae bloom dilution of Hg. It also relates to studies that have reported algae bloom dilution of As (Chen & Folt, 2000), and polychlorinated biphenyls (Larsson et al., 1992).

Two mechanisms may explain these findings. First is surface availability (Chen & Folt, 2005). This means the phytoplankton share a finite pool of metals and have a constant uptake. Thus enhanced lake productivity reduced the mass-specific metal concentrations. Yet it is difficult to accept that surface availability controlled metal uptake by the phytoplankton alone because the mass-specific concentrations of Mn showed no correlation with TP (r2 = 0.0004), while Co (and P) showed no significant decline with increasing TP concentrations. Secondly, because the trace element to macronutrient (i.e., phosphorus or carbon) ratios is a balance of net steady-state uptake and growth rates (Sunda & Huntsman, 1997; Sunda & Huntsman, 2004). As nutrients become more available, growth rates increase, which eventually results in a decline in element to phosphorus ratios in the cells. The significant correlations (p < 0.05) between the mass-specific metal (Pb, Cd, Cr, Hg) to P ratios in phytoplankton and TP (Fig. 5), and their negative correlation against chlorophyll-a appear to be in agreement with this biodilution hypothesis. This also may explain why Mn showed no correlation with TP. Mn is an essential element for phytoplankton growth (Morel, Hudson & Price, 1991), and so new cells may assimilate the available Mn.

Figure 9 indicates that the relationship of increasing TP and cell count is subject to exponential growth (Serruya & Berman, 1975). Figure 8 suggests the relationship of increasing cell numbers and their Hg, Pb, Cd, Cu, Co, Ni and Cr concentrations follows the pattern of exponential decay. The association between Figs. 8 and 9 provides potential insight into the rate at which algae bloom dilution occurs. That is, as TP increases, phytoplankton cell growth accelerates gently, and the concentration of metals in cells rapidly decline until it approaches zero, where the rate of the absolute decrease in the metals reduces. This deceleration in algae bloom dilution may eventually be paralleled by a lack of P to sustain the growth of more phytoplankton or insufficient growth space.

Table 6 Metal to P stoichiometries (mol:mol) of the freshwater phytoplankton collected in Loch Coire nan Arr, Loch Doilet and Loch Urr for this study.

Calculations were based on the mean concentrations of the metals per cell in the three lakes (Table 5). These were then converted to molar concentrations, and divided by the sum of all components, which included C and N molar concentrations that were calculated based on the standard Redfield (1958) ratio of C106:P1:N16. The averages of the ratios across the lakes yields a mean metal to P stoichiometry of (C106P1N16)1000Pb0.019Hg0.00004Cu0.013Cd0.005 Cr0.2Co0.0008Mn0.2Ni0.012.

Element	Loch Coire nan Arr	Loch Doilet	Loch Urr	Mean	
C	105,860	106,197	105,989	106,015	
N	15,979	16,030	15,998	16,002	
P	999	1,002	1,000	1,000	
Pb	0.03	0.01	0.01	0.019	
Hg	0.00005	0.00003	0.00003	0.00004	
Cu	0.02	0.01	0.01	0.013	
Cd	0.009	0.004	0.002	0.005	
Cr	0.3	0.1	0.1	0.2	
Co	0.001	0.001	0.001	0.0008	
Mn	0.3	0.1	0.3	0.2	
Ni	0.01	0.01	0.01	0.012	

The exponential relationships in Fig. 8 also suggest that the selective uptake of metals by the phytoplankton occurred (Santana-Casiano et al., 1995). If the decay constants in Fig. 8 are examined, it is evident that the rate of Pb decay in phytoplankton with increasing cell number is more rapid than Cu with respective decay constants of 0.0046 and 0.0037. It is also evident that algae bloom dilution is least effective on the most essential metal Mn with a decay constant of 0.0031. The differences in the rate constants of the algae bloom dilution suggest the involvement of two intracellular mechanisms in the selective uptake of metals. One is metabolic, which attempts to sustain the essential metals (e.g., Mn) concentrations (Sunda & Huntsman, 1998). The other is a detoxification process that stores excess P as intracellular polyphosphate, which protects the cells by binding with metals in a detoxified form (Walsh & Hunter, 1995). If the correlation between the ratios of metals to P in cells with TP in this study (Fig. 5) is consulted again, it is notable that the only metals that showed a significant decrease in their ratio to P were Pb, Cd, Hg and Cr. It is also notable that these four metals had a strikingly similar decay constant with their relationship in phytoplankton to increasing cells. That is, 0.0046 for both Pb and Cd, and 0.0045 for Cr and Hg (Fig. 8). Additionally, of the metals tested in this study, these four metals are considered the most toxic to phytoplankton (Xue & Sigg, 1993). Therefore, it is possible that when nutrients became more available, growth rates and cellular P increased, forming intracellular polyphosphate bodies that selected less toxic metals more rapidly.

Table 6 presents the metal to P stoichiometries (mol:mol) of the freshwater phytoplankton collected in this study. The calculations were based on the mean concentrations of the metals per cell in each of the three lakes (Table 5). These were converted to molar concentrations and divided by the sum of all components, which included the C and N molar concentrations based on the standard Redfield (1958) ratio of C106:P1:N16. Table 6 shows the ratios of the metals between the lakes are in the same order of magnitude. The mean metal to P stoichiometry from this investigation is (C106P1N16)1000Pb0.019Hg0.00004Cu0.013Cd0.005Cr0.2Co0.0008Mn0.2Ni0.012. This is similar to the phytoplankton cell stoichiometry presented by Twining, Baines & Fisher (2004) who found, for instance, 0.26 mol of Mn for every 1 mol of P, whereas this study found 0.21 mol of Mn for every 1 mol of P. The slightly higher ratio offered by Twining et al. may be expected as their study was on marine phytoplankton. This is because P is generally more concentrated in the phytoplankton of freshwater lakes, and thus lowering the metal to P ratio.

The calculated stoichiometry may be used to estimate the concentration of metals per phytoplankton cell in the lakes based on cell size. If the average biomass of one cell is 1.55 × 10−10 g, and using the Cd: P ratio of 0.000005/1, the estimated Cd concentration bound to a cell is 7.76 × 10−16 mol (or 87.2 × 10−15 g). If the P concentration is raised by a factor of 4, the estimated Cd is 3.11 × 10−18 mol (or 3.49 × 10−16 g). The risk of toxicity can then be predicted by comparing the results to those of toxicity tests. For instance, Wang & Dei (2006) observed toxicity at a Cd:P ratio of >0.2. While this may be useful, using the stoichiometry as a predictor on a wider scale than the lakes investigated has large uncertainties because it would assume the ratio is constant.

Conclusions

1. A higher TP concentration in the lakes resulted in significant algae growth dilution of the mass-specific Pb, Cd, Hg, Cu, Ni and Cr in the phytoplankton. This was because the available metals had to be shared among more and as P became more available, the mass specific metal to P ratios in the phytoplankton declined. The same mechanisms were not effective on Mn because it is assimilated during phytoplankton growth.

2. The relationship between the number of phytoplankton cells per millilitre of lake water and the mass-specific metal concentrations in the phytoplankton provides an examination of the rate of algae bloom dilution in the lakes. As TP increased, phytoplankton cell growth accelerated gradually, and the concentration of metals in cells rapidly declined until it approached zero. The decay constants indicate that Mn has the lengthiest rate of algae bloom dilution among the metals. This suggests the involvement of two intracellular mechanisms in the active selection of metals. The first is metabolic in that growing cells have preference for Mn and thus it is diluted at a more gradual rate. The second is a detoxification process that stores excess P as intracellular polyphosphate, which selects the less toxic metals more rapidly.

3. The simultaneous measurements of metals and P in phytoplankton cells, along with quantification of changes in cell mass, generated a mean metal to P stoichiometry of (C106P1N16)1000Pb0.019Hg0.00004Cu0.013Cd0.005Cr0.2Co0.0008Mn0.2Ni0.012 based on the Redfield average C, N and P stoichiometry of (CH2O)106(NH3)16H3PO4. This stoichiometry can be used to estimate the concentration of metals in cells based on their P content and may be incorporated into BLM if the concentration of cell surfaces were to be used as the biotic ligands.

Supplemental Information

Table S1 Quantitative concentrations of metals and P that showed linearity in the calibration curves

Fully quantitative concentrations of metals and P that showed linearity in the calibration curves computed by Plasmalab. These were subsequently used in the regression analysis to determine the concentration of the elements in the unknown sample solutions.

Click here for additional data file.

Additional Information and Declarations

Competing Interests

Author Contributions

Data Availability

1 Sulphur (S), potassium (K), magnesium (Mg), calcium (Ca), strontium (Sr), cobalt (Co), molybdenum (Mo).

The authors declare there are no competing interests.

Aine M. Gormley-Gallagher conceived and designed the experiments, performed the experiments, analyzed the data, contributed reagents/materials/analysis tools, wrote the paper, prepared figures and/or tables, reviewed drafts of the paper.

Richard W. Douglas and Brian Rippey contributed reagents/materials/analysis tools, reviewed drafts of the paper.

The following information was supplied regarding data availability:

EThOS, British Library: http://ethos.bl.uk/OrderDetails.do?uin=uk.bl.ethos.529532.

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
