# Peer review of "Metal to phosphorus stoichiometries for freshwater phytoplankton in three remote lakes"

_PeerJ, doi:10.7717/peerj.2749_

## Round 0.1 · original submission · Minor Revisions

I agreed with the reviewers that the paper mainly needs to be re-written for clarity. In general the results and conclusions appear valid and supported by the data. Please respond to all changes recommended by the reviewers.

Reviewer 1 ·

Basic reporting

The manuscript “Metal to phosphorus stoichiometry for freshwater phytoplankton in three Scottish lakes” addresses the changes in metal content relative to phosphorus across lakes with differing amounts of atmospheric deposition and trophic status. The manuscript clearly shows that a great deal of work was conducted in order to investigate this topic and the conclusions that are drawn form the body of work are well constructed and are founded and built upon a thorough review of the literature. Where the manuscript lacks is in the development of a clear hypothesis and in the introduction of why this work is needed beyond a short call to arms at the end of the introduction. There are several areas where more careful review of language and attention to detail are needed. Justification for methods and analysis are lacking and the figures appear slightly unprofessional. Reading the document this reviewer felt that there was an ebb and flow of preparation that gave the ms a feeling of unpolishedness. For instance, there were entire paragraphs that felt boisterous and superfluous while others where extremely well written, detailed, and apt. It is this reviewers opinion that the science and foundational concepts communicated in the ms are solid and important and worth of publication, however, without major review of the flow, language, and a strong editorial polishing this manuscript is not publishable as it is. A list of minor comments is in the General comments to Authors section. (It should be noted that minor comments should not be considered a check list of issues, but rather are intended to serve as guidance and should be applied to the rest of the document)
A very simple, yet substantial improvement that will help the overall tone of the document is to set up a list or series of hypotheses. This will allow for a more compact and patterned tone to the rest of the document. This could be as simple as taking the outline of the discussion (which is rather well written) and using the introduction to set up a hypothesis/specific aim for each component. Then tailor each section of the ms to this pattern/form and it will tidy up the document greatly.
The graphs appear unprofessional and need an overhaul. It is difficult to determine which axis apply to which data point, why curvilinear lines are used to connect the points rather than straight lines, the coloration when simple differences in symbol would suffice all add up to a rather limited feel. Considering the PeerJ’s commitment to ease to submission this point has limited value in the review and is simply a note to the authors.

Experimental design

One of the only critiques from a scientific stand point was the use of predictive models to estimate chl a from TP when you measured chl a directly. I think more explanation as to the logic behind this approach is needed as it feels as though this approach is not needed and clutters the figures and the interpretation of the results. Other than this the conclusions and data used to back them up where well done. Additionally, L148: this likely lead to size selection of smaller phytoplankton. Although the mesh was large enough for larger algae, as the mesh clogged with zoops, it likely was in reality much smaller the 250 UM. Was there any smaller than 250 phytoplankton found in the larger size fraction?

Validity of the findings

I found the interpretation and concluding statements the strong point o this paper.

Additional comments

Minor comments
The introduction needs polishing in order to get the reader ready for the data that is coming. As mentioned before hypotheses will help this. There are many grammatical and language mistakes that can be cleaned up quickly with an eye towards detail. Some areas to look for improvement follow:
L26: Are not all phytoplankton composed of C, N and P?
L30: Delete in the same context,
L33: micronutirent metals
L42: uptake not update
L46: Awkward… Perhaps… extended the concept of Redfield nutrient stoichiometry to.
L56: acknowledged that the stoichiometries are approximations that could vary...
L58: Delete a, ….calculated stiochometries
L59: based on regressions of P concentrations
L65: clarify what fish gills is referencing, I assume it is an active site for ligands but this should be stated
L84: Revise Sentence
L117: This sentence is awkward and needs revision. I am not sure what you are trying to say
L129: As written this is a 3 month period, do you mean 2007
L138: Standard methods?
L140: To concentrate phytoplankton not of concentrated phytoplankton
L150: Delete their
L151: Delete fixative
L154: simple state that the samples were preserved with GTA at a final concentration of 2% and delete this sentence
L162: measured spectrometrically after digestions at 882nm
L183: This whole paragraph is not needed. Just state that you calculated the surface pare of cell per L of lake water
L189: Revise this sentence Acid digestions we prepared using methods found in Reynolds
L212: No need for this table whatsoever
L252: This is a table not text. There is no need to bombard the reader with a list like this, please make a table and refer to trends or statistical descriptions in the results text, Change throughout results.
L288: The Hg plot should have the axis cross on the far left side of the plot crossing the y at -1.0
At this point I think you see what these changes are illustrating and I leave it up to authors to do a proper edit of the ms for clarity, brevity and flow.

Reviewer 2 ·

Basic reporting

Good introduction reviewing the progression of the science of metal dynamics and stoichiometry as they relate to toxicity and phytoplankton growth.

Line 42
Should “update” be changed to “uptake”?

Line 215
Table 2. Does not need to be in the main body. Suggest adding it to the support documentation.

Lines 235-237
I would replace Figures 1A-1C with one figure that depicted the relationship of chlorophyll a (y-axis) and total phosphorus (x-axis) for all three lakes. I believe this would present the data better and allow for comparison across all three lakes. It would depict the differences in productivity among the three lakes. If the authors choose to retain the time-series figures I would suggest removing the data series generated by the two TP/Chl a models (OEDC 1982, Prairie et al. 1989) as those relationships are not used in any of the analyses in this manuscript.

Lines 265-276
In Figures 2A-B, please explain the negative values for lead (Pb) concentration of metal per unit mass of phytoplankton.

Line 289
Lower, right figure needs to be labeled “Ni”?

Line 327
Figures should not have negative number for the y-axis minimum.

Lines 368-372
Can Figures 6A-C be removed from the manuscript. The time-series do not seem to add to the results and the same phytoplankton metrics are reported in Figures 7A-C.

Lines 385-388
For Figures 7A-C, should total phosphorus (TP) concentration be the independent variable (x-axis) and the phytoplankton metrics be the dependent variables (y-axis)?

Line 403
Per the comment above (Lines 385-388) the # of cells per ml is estimated by TP concentration. TP is independent variable and # of cells per ml is the dependent variable

Line 430
Figure 4C should be replaced with Figure 7C.

Lines 520-562
This section of the discussion related to the variability of TP and chlorophyll a is long and does not add to the manuscript.

Line 558
Also, phytoplankton <20 um is also not accounted for in this study.

Lines 563 and 570
Figure 6 should be replaced with Figure 7.

Experimental design

Lines 137-159
There is one issue related to the size-fractioning of phytoplankton in this paper that the authors should account for/address in the manuscript. A whole-water sample was used to identify and estimate biomass of phytoplankton. Net hauls were used to concentrate phytoplankton for metals analysis. The net-hauls had a mesh size of 20 µm. Thus, phytoplankton <20 µm were not part of the metals estimate. That is a potential source of error affecting the relationships of phytoplankton metals concentration/biomass and total phosphorus concentrations from lake water samples. Did the authors account for this difference in their analysis?

The authors described that phytoplankton was estimated from cell counts and cell sizes from the whole-water sample. But metals concentrations were measured from the net haul samples. How did the authors calculate the metals concentration per unit mass of phytoplankton?

The authors collected phytoplankton for metals analysis using a modified plankton net that resulted in a size range of organisms from 20-250 µm. Given my experience with size-fractioning zooplankton samples, I am surprised the authors did not find rotifers and copepod nauplii in the 20-250 µm fraction. Do these lakes have low zooplankton production?

Validity of the findings

The authors followed a logical progression through the manuscript and their results and analyses are valid. The results as related to the dilution of metals/biomass as a function of increased growth are consistent with other research findings (Karimi et al. 2007; Ward et al. 2010). The equations developed to estimate changes in metals concentration per biomass of phytoplankton over a range of lake phosphorus concentrations were based on statistically significant relationships. The metals concentration per unit mass of phytoplankton results in this manuscript are in the same range reported by Kuwabara et al. (2006).



CITATIONS
Karimi R, Chen CY, Pickhardt PC, Fisher NS, Folt CL. 2007. Stoichiometric controls of mercury dilution by growth. Proc Natl Acad Sci U S A. 2007 May 1; 104(18):7477-82.
Kuwabara, J.S., Topping, B.R., Woods, P.F., Carter, J.L., and Hager, S.W., 2006, Interactive effects of dissolved zinc and orthophosphate on phytoplankton from Coeur d'Alene Lake, Idaho: U.S. Geological Survey Scientific Investigations Report 2006-1034, 47 p. (Internet access at: http://pubs.usgs.gov/sir/2006/5091)

Ward, D. M., Nislow, K. H., Chen, C. Y., & Folt, C. L. (2010). Rapid, efficient growth reduces mercury concentrations in stream-dwelling Atlantic salmon. Transactions of the American Fisheries Society, 139(1), 1–10. doi:10.1577/T09-032.1

---

## Round 0.2 · Minor Revisions

Please consider the remaining comments of Reviewer 1 on Figure 3. Otherwise your paper is in good shape.

Reviewer 1 ·

Basic reporting

No Comments

Experimental design

No Comments

Validity of the findings

No Comments

Additional comments

The response of the authors to my previous review is adequate for publication. Several of my previous points were handled extremely well and I applaud the effort of the authors. The only area that I feel must be revised further is Figure 3. I think that it is far too busy and difficult to interpret. I understand the desire to use 2 y axis scales to show the temporal trends between all elements, however, it requires too much effort on the readers part and takes away form the value of the work. A six panel plot with the high concentrations on one side and the low concentrations on the other will convey the same message and be much easier to understand.

Reviewer 2 ·

Basic reporting

The authors responded to my review questions and concerns and made significant improvements to the manuscript. This manuscript should be accepted for publication.

Experimental design

No Comments

Validity of the findings

No Comments

---

## Round 0.3 · accepted · Accept

Your paper is now in fine shape for publication. Thanks for hanging in there on getting this one done. It is a nice contribution.